# LEARNING TO SELECT EXAMPLES FOR PROGRAM SYNTHESIS

## ABSTRACT

Program synthesis is a class of regression problems where one seeks a solution, in the form of a source-code program, that maps the inputs to their corresponding outputs exactly. Due to its precise and combinatorial nature, it is commonly formulated as a constraint satisfaction problem, where input-output examples are encoded as constraints and solved with a constraint solver. A key challenge of this formulation is scalability: while constraint solvers work well with few well-chosen examples, constraining on the entire set of examples incurs significant overhead in both time and memory. In this paper we address this challenge by constructing a representative subset of examples that is both small and able to constrain the solver sufficiently. We build the subset one example at a time, using a trained discriminator to predict the probability of unchosen input-output examples conditioned on the chosen input-output examples, adding the least probable example to the subset. Experiment on a diagram drawing domain shows our approach produces subsets of examples that are small and representative for the constraint solver.

## 1 INTRODUCTION

Program synthesis (or synthesis for short) is a special class of regression problems where rather than minimizing the error on an example dataset, one seeks an exact fit of the examples in the form of a program. Applications include synthesizing database relations (Singh et al. (2017)), inferring excel-formulas (Gulwani et al. (2012)), and compilation (Phothilimthana et al. (2016)). In these domains, the synthesizer was able to come up with complex programs consisting of branches, loops, and other programming constructs. Recent efforts (Ellis et al. (2015); Singh et al. (2017)) show an interest in applying the synthesis technique to large sets of examples, but scalability remains an open problem. In this paper we present a technique to select from a large dataset of examples a representative subset that is sufficient to synthesize a correct program, yet small enough to solve efficiently.

There are two key ingredients to a synthesis problem: a domain specific language (DSL for short) and a specification. The DSL defines a space of candidate programs which serve as the model class. The specification is commonly expressed as a set of example input-output pairs which the candidate program needs to fit exactly. The DSL restricts the structure of the programs in such a way that it is difficult to fit the input-output examples in an ad-hoc fashion: This structure aids generalization to an unseen input despite "over" fitting the input-output examples during training.

Given the precise and combinatorial nature of a synthesis problem, gradient-descent based approaches perform poorly and an explicit search over the solution space is required (Gaunt et al. (2016)). For this reason, synthesis is commonly casted as a constraint satisfaction problem (CSP) (Solar-Lezama (2013); Jha et al. (2010)). In such a setting, the DSL and its execution can be thought of as a parametrized function $F$, which is encoded as a logical formula. Its free variables $s \in S$ correspond to different parametrization within the DSL, and the input-output examples $D$ are expressed as constraints which the instantiated program needs to satisfy, namely, producing the correct output on a given input.

$$\exists s \in S. \bigwedge_{(x_i, y_i) \in D} F(x_i; s) = y_i \, .$$

The encoded formula is then given to a constraint solver such as Z3 (de Moura & Bjørner (2008)), which solves the constraint problem, producing a set of valid parameter values for $s$. These values are then used to instantiate the DSL into a concrete, executable program.

**A key challenge**    of framing a synthesis problem as a CSP is that of scalability. While solvers have powerful built-in heuristics to efficiently prune and search the constrained search space, constructing and maintaining the symbolic formula over a large number of constraints constitute a significant overhead. For this reason, significant efforts were put into simplifying and re-writing the constraint formula for a compact representation ( Singh & Solar-Lezama (2016); Cadar et al. (2008)). Without such optimizations, it is possible for a formula to exceed the computer's memory. If one wishes to apply program synthesis to a sufficiently large dataset, there needs to be a way to limit the number of examples expressed as constraints.

The standard procedure to limit the number of examples is counter example guided inductive synthesis, or CEGIS for short (Solar-Lezama et al. (2006)). CEGIS solves the synthesis problem with two adversarial sub-routines, a synthesizer and a checker. The synthesizer solves the CSP with a subset of examples rather than the whole set, producing a candidate program. The checker takes the candidate program and produces an adversarial counter example that invalidates the candidate program. This adversarial example is then added to the subset of examples, prompting the synthesizer to improve. CEGIS successfully terminates when the checker fails to produce an adversarial example. By iteratively adding counter examples to the subset, CEGIS can drastically reduces the size of the constraint constructed by the synthesizer, making it scalable to large domains. The subset of examples are *representative* in a sense that, once a candidate program is found over this subset, it is also correct over all the examples. However, CEGIS has to repeatedly invoke the constraint solver in the synthesis sub-routine to construct the subset, solving a sequence of challenging CSP problems. Moreover, due to the phase transition Gent & Walsh (1994) property of SAT formulas, there may be instances in the sequence of CSP problems where there are enough constraints to make the problem non-trivial, but not enough constraints for the solver to properly prune the search space[1], causing the performance of CEGIS to become extremely volatile.

In this paper, we construct the *representative subset* in a different way. Rather than using the constraint solver as in CEGIS, we directly learn the relationships between the input-output examples with a neural network. Given a (potentially empty) subset of examples, the neural network computes the probability for other examples not in the subset, and grow the subset with the most "surprising" example (one with the smallest probability). The reason being if an input-output example has a low probability conditioned on the given subset, then it is the most constraining example that can maximally prune the search space once added. We greedily add examples until all the input-output examples in the dataset have a sufficiently high probability (no longer surprising). The resulting subset of examples is then given to the constraint solver. Experiments show that the trained neural network is capable of representing domain-specific relationships between the examples, and, while lacking the combinatorial precision of a constraint solver, can nonetheless finds subset of representative examples. Experiment shows that our approach constructs the sufficient subset at a much cheaper computational cost and shows improvement over CEGIS in both solution time and stability.

## 2    AN EXAMPLE SYNTHESIS PROBLEM

To best illustrate the synthesis problem and the salient features of our approach, consider a diagram drawing DSL (Ellis et al., 2017) that allows a user to draw squares and lines. The DSL defines a $draw(row, col)$ function, which maps a $(row, col)$ pixel-coordinate to a boolean value indicating whether the specified pixel coordinate is contained within one of the shapes. By calling the $draw$ function across a canvas, one obtains a rendering of the image where a pixel coordinate is colored white if it is contained in one of the shapes, and black otherwise. Figure 1 shows an example of a draw function and its generated rendering on a 32 by 32 pixel grid. The drawing DSL defines a set of parameters that allows the $draw$ function to express different diagrams, some of which are underlined in Figure 1(left). The synthesis problem is: Given a diagram rendered in pixels, discover the hidden parameter values in the draw function so that it can reproduce the same rendering.

---

[1]Imagine a mostly empty Sudoku puzzle, the first few numbers and the last few numbers are easy to fill, whereas the intermediate set of numbers are the most challenging

```
1  def draw(row, col):
2      # shape constructor
3      shapes = []
4      for i in range(3):
5          for j in range(3):
6              offset_x = 10*i + 0*j + 5
7              offset_y = 0*i + 10*j + 0
8              s = square(0+offset_x, 5+offset_y)
9              l1 = line(0+offset_x, 2+offset_y, 0+offset_x,
10                       -7+offset_y, False, True, True)
11             l2 = line(0+offset_x, 5+offset_y, -7+offset_x,
12                       5+offset_y, True, False, True)
13             shapes += [s, l1, l2]
14     # inclusion check
15     for s in shapes:
16         if inside(s, row, col):
17             return True
18     return False
```

Figure 1: Sketch of code to draw an image (left) and the generated image (right). Boxes are drawn around the many adjustable parameters in the code such as the number of iterations and offsets for the shapes.

The synthesized drawing program is correct when its rendered image matches the target rendering exactly. Let $Sdraw$ be the synthesized draw function and $Target$ be the target rendering:

$$\textbf{correct}(Sdraw) \coloneqq \forall(row, col).\ Sdraw(row, col) = Target[row][col]$$

Because of the many possible combinations of parameters for the program, this is a difficult combinatorial problem that requires the use of a constraint solver. Each of the pixel in the target render is encoded as an input-output pair $((row, col), bool)$, which can be used to generate a distinct constraint on all of the parameters. For the 32 by 32 pixel image, a total of 1024 distinct constraints are generated, which impose a significant encoding overhead for the constraint solver.

In this paper, we propose a algorithm that outputs a representative subset of input-output examples. This subset is small, which alleviates the expensive encoding overhead, yet remains representative of all the examples so that it is sufficient to constrain the parameter only on the subset. Figure 2 (left) shows the selected subset of examples: white and black pixels indicate chosen examples, grey pixels indicate unchosen ones. As we can see, from a total of 1024 examples, only 15% are selected for the representative subset. The representative subset is then given to the constraint solver, recovering the hidden parameter values in Figure 2 (right).

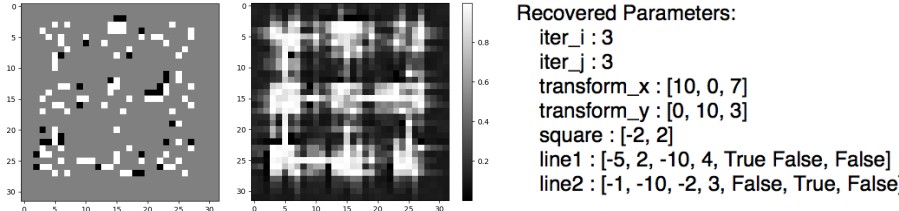

Figure 2: The selected subset of pixel examples (left). The neural network's estimation of the image (middle). The recovered parameters from running the solver on the selected subset (right).

The algorithm constructs the representative subset iteratively. Starting with an empty subset, the algorithm uses a neural network model to compute the probability of all the examples conditioned on the chosen examples in the subset. It then adds to the subset the least probable example, the intuition being the example with the lowest probability would best restrict the space of possible solutions. The process stops when all the examples in the dataset are given a sufficiently high probability. In the context of the drawing DSL, the process stops when the neural network is sufficiently confident in its reconstruction of the target rendering given the chosen subset of pixels Figure 2 (middle). The rest of the paper elaborates the specifics of our approach.

## 3 EXAMPLES REDUCTION

The crux of our algorithm is an example selection scheme, which takes in the set of examples and outputs a small subset of representative examples. Let D' $\subseteq$ D be a subset of examples. Abusing notation, let us define the *consistency constraint* D'$(s) := \bigwedge_{(x_i, y_i) \in D'} F(x_i; s) = y_i$, that is to say, the parameter $s$ is consistent with all examples in D'. We define the *smallest sufficient subset* as:

$$D^* = \underset{D' \subseteq D}{\operatorname{argmin}} |D'| \ \ s.t. \ \forall s \in S. \ D'(s) \Rightarrow D(s).$$

$D^*$ is *sufficient* in a sense any parameter $s$ satisfying the subset $D^*$ must also satisfy D. Finding the exact minimum sized $D^*$ is intractable in practice, thus we focus on finding a sufficient subset that is as close in size to $D^*$ as possible.

### 3.1 EXAMPLES REDUCTION WITH A COUNT ORACLE

In this subsection we describe an approximate algorithm with a count oracle $c$, which counts the number of valid solutions to a subset of examples: $c(D') := |\{s \in S | D'(s)\}|$. This algorithm constructs the subset D' greedily, choosing the example that maximally restricts the solution space.

D' = {}
**while** *True* **do**
    $(x_i, y_i) \leftarrow \operatorname{argmin}_{x_j, y_j} c(D' \cup \{(x_j, y_j)\})$ # selection criteria
    **if** $c(D') = c(D' \cup \{(x_i, y_i)\})$ **then**
       | **return:** D'
    **else**
       | D' $\leftarrow$ D' $\cup \{(x_i, y_i)\}$
    **end**
**end**

**Algorithm 1:** An example reducing algorithm with a count oracle

**Claim 1:** Algorithm 1 produces a subset D' that is sufficient, i.e. $\forall s \, D'(s) \Rightarrow D(s)$.

**Proof 1:** The termination condition for Algorithm 1 occurs when adding any example to D', the counts remain unchanged $c(D') = c(D' \cup \{(x, y)\}), \forall (x, y) \in D$. As D'$(s)$ is defined as a conjunction of satisfying each example, $c$ can only be monotonically decreasing with each additional example, as more solutions become invalidated: $c(D') \geq c(D' \cup \{(x, y)\})$. At termination, equality occurs for every additional $(x, y) \in D$, where no more solutions are invalidated, thus we have the sufficiency condition $\forall s. \, D'(s) \Rightarrow D(s)$.

**Claim 2:** Algorithm 1 produces a subset D' that is $1 - \frac{1}{e}$ optimal

**Proof Gist:** To show this, we need to show the count function $c(D')$ is both monotonic and submodular (Nemhauser et al., 1978). We have already shown monotonicity in the previous proof, for the sub-modularity proof see appendix.

To use Algorithm 1, one needs to solve the model counting problem (Gomes et al., 2008) for the count function $c$, which is impractical in practice. We now aim to resolve this issue by adopting an alternative selection criteria.

### 3.2 EXAMPLE SELECTION WITHOUT THE COUNT ORACLE

The selection criteria in Algorithm 1 uses the count oracle $c$, which is impractical to compute in practice, in this sub-section, we develop an alternative selection criteria that can be approximated efficiently with a neural network. Let D' $= \{(x^{(1)}, y^{(1)}) \ldots (x^{(r)}, y^{(r)})\}$ where $(x^{(j)}, y^{(j)})$ denotes the $j^{th}$ input-output example to be added to D'. We define the *selection probability*:

$$Pr((x, y) | \text{D'}) := Pr(F(x; s) = y | \text{D'}(s))$$

Note that $Pr((x, y) | \text{D'})$ is **not** a joint distribution on the input-output pair $(x, y)$, but rather the probability for the event where the parameterized function $F(\cdot; s)$ maps the input $x$ to $y$, conditioned on the event where $F(\cdot; s)$ is consistent with all the input-output examples in D'. We will now show that one can use $Pr((x, y) | \text{D'})$ as the selection criteria rather than the count oracle in Algorithm 1.

**Claim:** Under a uniform distribution of parameters $s \sim unif(S)$,

$$\underset{(x,y)}{\operatorname{argmin}} \, c(\text{D'} \cup \{(x, y)\}) = \underset{(x,y)}{\operatorname{argmin}} \, Pr((x, y) | \text{D'})$$

**Proof:** See appendix.

To use $\operatorname{argmin}_{(x,y)} Pr((x, y) | \text{D'})$ as a selection criteria to grow the subset D', one needs a corresponding termination condition. It is easy to see the right termination condition should be $\min_{(x,y)} Pr((x, y) | \text{D'}) = 1$: when all the input-output examples are completely determined given D', the subset is sufficient.

### 3.3 APPROXIMATING SELECTION PROBABILITY WITH A NEURAL NETWORK

We now describe how to model $Pr((x, y) | \text{D'})$ with a neural network. For the scope of this work, we assume there exists an uniform sampler $s \sim unif(S)$ for the possible parameters, and that the space of possible input and output values are finite and enumerable $dom(x) = \dot{x}_1 \ldots \dot{x}_N$, $dom(y) = \dot{y}_1 \ldots \dot{y}_M$. We will first describe a count-based approach to approximate $Pr((x, y) | \text{D'})$, then describe how to model it with a neural network to achieve generalization properties.

For the count-based approximation, we sample a subset of input values $X' = \{x^{(1)}, \ldots, x^{(r)}\}$, and a particular input value $x \notin X'$. We sample a parameter $s \in S$ and evaluate the parameterized function, $F(\cdot; s)$, on each of the input in $X'$, obtaining output values $F(x^{(1)}; s) = y^{(1)}, \ldots, F(x^{(r)}; s) = y^{(r)}$, we also evaluate the function on $x$, obtaining $F(x; s) = y$. Let $\hat{c}$ denote the empirical count, we have, after sufficient number of samples:

$$Pr((x, y) | \text{D'}) \approx \frac{\hat{c}(F(x^{(1)}; s) = y^{(1)}, \ldots, F(x^{(r)}; s) = y^{(r)}, F(x; s) = y)}{\hat{c}(F(x^{(1)}; s) = y^{(1)}, \ldots, F(x^{(r)}; s) = y^{(r)})} \, .$$

The issue with the count-based approach is that we need sufficient samples for any subset of inputs, with a total number of $2^N$ subsets where $N = |dom(x)|$. Therefore, we approximate $Pr((x, y) | \text{D'})$ with a neural network.

The neural network is set-up similarly to a feed-forward auto-encoder with $N$ input neurons $\mathcal{Y}_1 \ldots \mathcal{Y}_N$ and $N$ output neurons $\mathcal{Y'}_1 \ldots \mathcal{Y'}_N$. That is to say, we enumerate over (the finite set of) distinct input values $\dot{x}_1 \ldots \dot{x}_N$, creating a corresponding input and output neuron each time. Each input neuron $\mathcal{Y}_i$ can take on $1 + M$ different values where $M = |dom(y)|$, and each output neuron $\mathcal{Y'}_i$ can take on $M$ different values, both assume a 1-hot encoding. In this encoding, each input neuron $\mathcal{Y}_i$ and output neuron $\mathcal{Y'}_i$ can represent the value of running function $F(\cdot; s)$ on the corresponding input value $\dot{x}_i$, $F(\dot{x}_i; s)$. The value $F(\dot{x}_i; s) \in dom(y)$ is represented as a distinct class in $1 \ldots M$. Input neuron $\mathcal{Y}_i$ can also represent an additional class, $M + 1$, representing the unknown value. Figure blah shows our neural network architecture, note that we do not suggest a specific neural network architecture for the middle layers, one should select whichever architecture that is appropriate for the domain at hand. During training time, given a particular sampled parameter $s$ and a sampled subset of inputs $X' = \{x^{(1)}, \ldots, x^{(r)}\}$, we set the input and output neurons values as follows:

$$\mathcal{Y}_i = \begin{cases} F(x_i, s) & \text{if } x_i \in X' \\ M + 1 & \text{otherwise} \end{cases} \qquad \mathcal{Y'}_i = F(x_i, s)$$

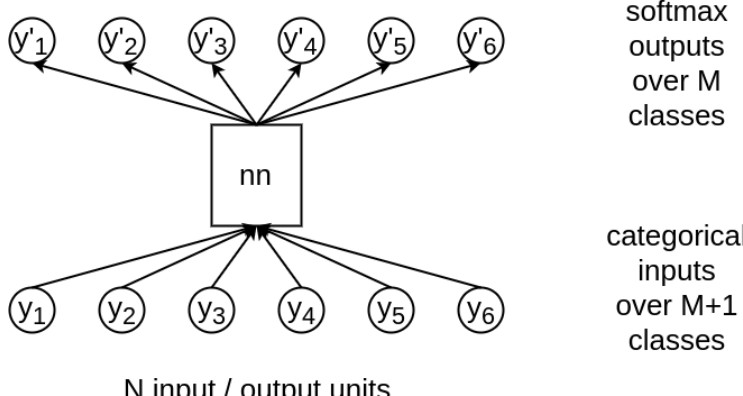

Figure 3: Our neural network architecture resembles a feed-forward auto-encoder with explicitly enumerated input and output neurons

That is to say, the training task of the neural network is to predict the output values for all the possible input values in $dom(x)$ while given only a subset of input-output values for D' = $\{(x^{(i)}, F(x^{(i)}; s)) | x^{(i)} \in X'\}$, the rest (where $x' \notin X'$) are encoded as unknowns. This is similar to a data completion task in Boltzmann machines (Ackley et al., 1985), with the difference that we directly compute the completion process rather than performing a gradient search for the most probable completion configuration.

During use time, given a subset of input-output examples D' = $\{(x^{(1)}, y^{(1)}) \ldots (x^{(r)}, y^{(r)})\}$, we set for each $x^{(i)}$ its corresponding input neuron with value $y^{(i)}$, and set the value unknown for neurons whose corresponding input values that do not occur in D'. The neural network then computes the softmax values for all the $M$ classes in all the output neurons, obtaining $Pr((x, y) | \text{D'})$ for every possible input-output examples simultaneously.

### 3.4 TYING UP THE LOOSE ENDS WITH CEGIS

In the previous subsections we described an examples reduction algorithm that builds the sufficient subset one example at a time by greedily selecting the lease likely input-output example given the examples in the subset. We also showed how one can approximate the selection probability $Pr((x, y) | \text{D'})$ by training an auto-encoder like neural network. The remaining problem lies in the approximate nature of the neural network: It cannot perfectly model the probability $Pr((x, y) | \text{D'})$, and thus we need to use a different termination condition for our example reduction algorithm. Rather than terminating the algorithm when $\min_{(x,y)} Pr((x, y) | \text{D'}) = 1$, we adopt a weaker termination condition $\mathbf{mean}_{(x,y)} Pr((x, y) | \text{D'}) \geq \beta$, terminating when the average probability of all the examples are greater than a certain threshold $\beta$.

By approximating the selection probability and relaxing the termination condition, one can no longer guarantee that the subset produced by our reduction algorithm is *sufficient*. That is to say, there may be solutions $s$ which satisfies the subset D' yet fails to satisfy the entire set of examples D. We can remedy this problem by leveraging CEGIS, which guarantees a solution $s$ that is correct on all the examples in D.

Like Algorithm 1, CEGIS also maintains a subset of examples D' and grows it one at a time. The difference being the selection criteria and termination condition. In CEGIS, two subroutines, synthesize and check, interacts in an adversarial manner to select the next example to add to the subset: The routine *synthesize* uses a constraint solver to produce a candidate parameter $s$ that satisfies the current D'; The routine *check* checks the candidate $s$ against all the examples in D, and finds a *counter example* $(x_{counter}, y_{counter})$ that invalidates the candidate $s$. This counter example is added to D', prompting the synthesizer to improve its solution. CEGIS terminates when no counter example can be found. Clearly, when CEGIS terminates, the resulting solution $s$ is correct on all the examples in D. By using a constraint solver in the synthesis step, and using the checker that checks against all the examples, CEGIS guarantees pruning of the solution space with each counter-example it adds to

```
D' = {}
while True do
    s = synthesize(S, D')
    (x_counter, y_counter) = check(s, D)
    if (x_counter, y_counter) == None then
        return: D'
    else
        D' = D' ∪ {(x_counter, y_counter)}
    end
end
```

**Algorithm 2:** CEGIS

the subset. The main drawback of CEGIS is that it requires repeated calls to the constraint solver in the synthesis step and no guarantees on how well the additional counter-example prunes the search space other than it invalidates the current candidate solution $s$.

**Our synthesis algorithm** combines the example selections and CEGIS. First, example selection is run until the mean selection probability reaches a certain threshold $\beta$, then the resulting set of sampled examples are given to CEGIS as the starting set of counter examples. CEGIS then repeatedly calls the constraint solver for candidate solutions, and checking each candidate solution against the entire example set $D$ until a correct solution is found:

```
# phase 1: examples selection
D' = {}
while mean_{(x,y)∈D} Pr((x,y)|D') ≤ β do
    (x,y) ← argmin_{x',y'} Pr((x',y')|D') # selection criteria
    D' ← D' ∪ {(x_i, y_i)}
end
# phase 2: CEGIS
while True do
    s = synthesize(S, D')
    (x_counter, y_counter) = check(s, D)
    if (x_counter, y_counter) == None then
        return: s
    else
        D' = D' ∪ {(x_counter, y_counter)}
    end
end
```

**Algorithm 3:** Synthesis with example selections

By initializing CEGIS with a set of representative examples, CEGIS will be able to find the correct solution with fewer calls to the constraint solver, saving both overhead time and solving time.

## 4 EXPERIMENTS

We perform a set of experiments measuring the overall speed and stability of our synthesis algorithm, and the representativeness of the subset of examples produced by the selection process. We evaluate our algorithm against 400 randomly generated images. For the experiment, the drawing function contains parameters that can generate a total of $1.31 \times 10^{23}$ possible programs. For each randomly generated image, the following synthesis algorithms are run:

- full: all 1024 examples are added to the subset, solved once
- rand: 10% of all examples are added to the subset, solved once
- nn: the subset generated by our selection algorithm, solved once
- CEGIS: the CEGIS algorithm where the check function return counterexamples in order

- rCEGIS: the CEGIS algorithm where the check function return counterexamples at random
- rand+CEGIS: initialize CEGIS with a random subset of 10% examples
- ours: our synthesis algorithm described in Algorithm 3, initializing CEGIS with the subset produced by the selection algorithm

Figure 4 shows the average time breakdown, the median and variance, and the number of examples selected for the different algorithms. Here, rand and nn are excluded because they are not guaranteed to synthesize a program that can perfectly reproduce the target render. On average, rand synthesizes a program that misses 10.1% of the pixels while nn misses 1.2%.

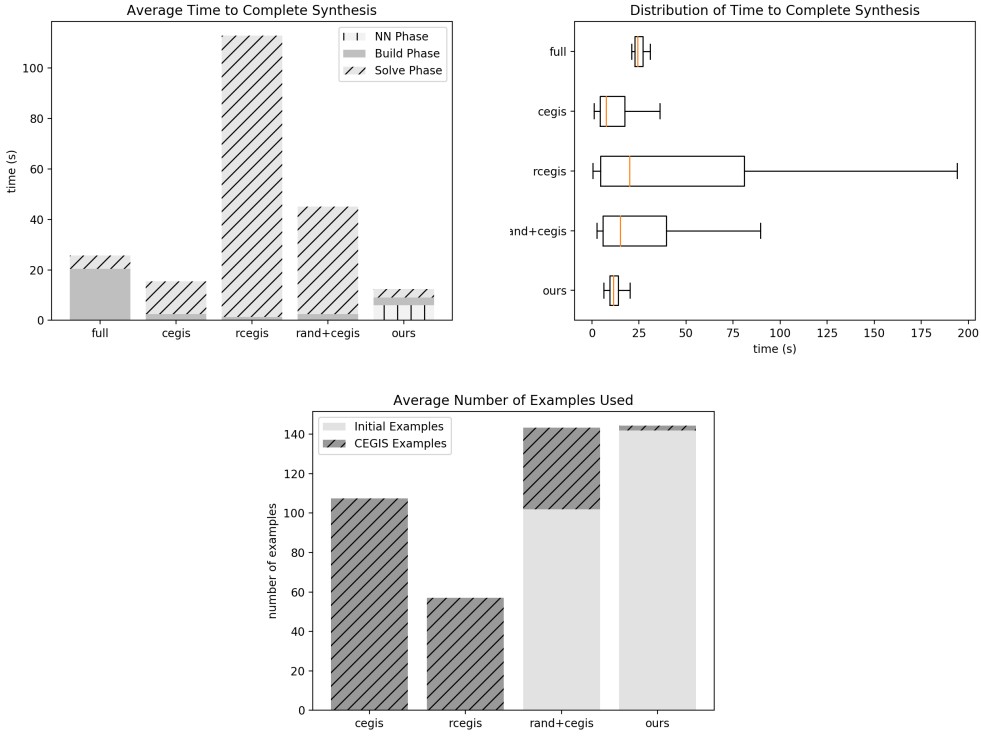

Figure 4: The average time taken for each step of the algorithm (upper left). The spread of total time taken for each algorithm (upper right). The number of examples used in each algorithm (bottom).

For the average time plot in Figure 4 (top left), we measure the breakdown for the different kinds of times: grey denotes the overhead time in constructing the constraints, slanted-stripes denotes the solving time, and vertical stripes denotes the time taken by the example selection algorithm. On average, our algorithm finishes the fastest, with cegis a close second. We remark that we achieve a similar solve time as the full algorithm, indicating the subset returned by our algorithm constrained the solver to a similar degree as constraining all the examples at once. In comparison, all the other algorithms have significantly longer solving time and shorter building time, indicating that these algorithms tend to under-constrain the synthesis problem, making it more difficult to solve. The drawback of our approach is the time it takes to produce the representative subset, which is around 6 seconds. This constitute as another form of overhead cost, but as we can see compared to the overhead cost of constructing the constraint, it is justified.

For the median and variance plot in Figure 4 (top right) for average over-all time, we note cegis comes in first for smallest median time, while our algorithm is second place. However, we remark that our algorithm has a much smaller variance in over-all time, achieving higher stability than cegis. One salient feature of this plot is that although cegis and rcegis only differs by which counterexample is added to the subset (the "first" one versus a random one), this small difference results in a huge difference in the over-all time performance. We postulate that cegis is able to leverage

the particular ordering of choosing the counter-examples are top-left to bottom-right, which tend to produce representative examples in the drawing DSL domain we have considered. By removing this particular ordering of choosing counter-examples, rcegis incurs a significant increase in solving time.

For the number of examples plot in Figure 4 (bottom), we measure the average number of examples in the selected subset. For this plot, solid grey measures the size of the initial subset of examples, and stripped measures additional examples chosen by the cegis algorithm. We note that rcegis on average was able to solve the synthesis problem with the least number of examples. However, rcegis also performs the worst in term of over-all solving time, suggesting that while it is possible to generate a valid solution from a small subset of examples, it is likely the case where that subset is not sufficiently constraining so the solver cannot efficiently prune the search space. By comparison, both cegis and our approach, the top two performing algorithms in overall-time, selects many more examples for the representative subset. We note that although rand+cegis selects roughly 70% of the examples as our approach, the subset it selected are not representative. This is evident in the high over-all solving time for rand+cegis, especially in solving time, indicating the examples it selected are not constraining the solver in a helpful way. By contrast, the subset selected by our selection algorithm is almost perfect, with only 1.5 additional counter-examples needed from CEGIS to arrive at a correct solution that matches all the pixels.

Overall, our algorithm provides a quick and stable solution over existing algorithms, and the subset that it provides is small and representative of the whole subset.

## 5 RELATED WORKS

In recent years there have been an increased interest in *program induction.* Graves et al. (2014), Reed & De Freitas (2015), Neelakantan et al. (2015) assume a differentiable programming model and learn the operations of the program end-to-end using gradient descent. In contrast, in our work we assume a non-differentiable programming model, allowing us to use expressive program constructs without having to define their differentiable counter parts. Works such as (Reed & De Freitas, 2015) and (Cai et al., 2017) assume strong supervision in the form of complete execution traces, specifying a sequence of exact instructions to execute, while in our work we only assume labeled input-output pairs to the program, without any trace information.

Parisotto et al. (2016) and Balog et al. (2016) learn relationships between the input-output examples and the syntactic structures of the program that generated these examples. When given a set of input-outputs, these approach use the learned relationships to prune the search space by restricting the syntactic forms of the candidate programs. In these approaches, the learned relationship is across the semantic domain (input-output) and the syntactic domain. In contrast, in our approach we learn a relationship between the input-output examples, a relationship entirely in the semantic domain. In this sense, these approaches are complimentary.

### ACKNOWLEDGMENTS

Use unnumbered third level headings for the acknowledgments. All acknowledgments, including those to funding agencies, go at the end of the paper.

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

APPENDIX

**Claim:** Algorithm 1 produces a subset D' that is $1 - \frac{1}{e}$ optimal

**Proof:** To show this, we need to show the count function $c(\text{D'})$ is both monotonic and sub-modular (Nemhauser et al., 1978). We have already shown monotonicity. For sub-modularity, we need to show for subsets $A \subset B \subseteq D$:

$$A \subseteq B \Rightarrow \forall (x,y) \in D. \; c(A) - c(A \cup \{(x,y)\}) \geq c(B) - c(B \cup \{(x,y)\})$$

To show this, we need to show the number of parameters $s$ invalidated by $(x,y)$ is greater in $A$ than that in $B$. Let $A'(s) := A(s) \wedge \neg\{(x,y)\}(s)$, the constraint stating that a parameter $s$ should satisfy $A$, but fails to satisfy $(x,y)$, similarly, let $B(s)' := B(s) \wedge \neg\{(x,y)\}(s)$. The count $c(A')$ indicates how many parameter $s$ becomes invalidated by introducing $(x,y)$ to $A$, i.e. $c(A') = c(A) - c(A \cup \{(x,y)\})$, similarly, $c(B') = c(B) - c(B \cup \{(x,y)\})$. Note that $A'$ and $B'$ are strictly conjunctive constraints, with $B'$ strictly more constrained than $A'$ due to $A \subseteq B$. Thus, there are more solutions to $A'$ than there are to $B'$, i.e. $c(A') \geq c(B')$, showing sub-modularity.

**Claim:** Under a uniform distribution of parameters $s \sim unif(S)$,

$$\operatorname*{argmin}_{(x,y)} c(\text{D'} \cup \{(x,y)\}) = \operatorname*{argmin}_{(x,y)} Pr((x,y)|\,\text{D'})$$

**Proof:** The probability $Pr((x,y)|\,\text{D'})$ can be written as a summation over all the possible parameter values for $s$:

$$
\begin{aligned}
Pr((x,y)|\,\text{D'}) :=& Pr(F(x;s) = y|\,\text{D'}(s)) \\
=& \sum_{s \in S} Pr(s|\,\text{D'}(s)) Pr(F(x;s) = y|s) \;.
\end{aligned}
$$

Note that under $s \sim unif(S)$, we have:

$$Pr(s|\,\text{D'}(s)) = \begin{cases} \frac{1}{c(\text{D'})} & \text{if D'}(s) \\ 0 & \text{otherwise} \end{cases} \;.$$

And since $F(\cdot\,;s)$ is a function we have:

$$Pr(F(x;s) = y|s) = \begin{cases} 1 & \text{if } F(x;s) = y \\ 0 & \text{otherwise} \end{cases} \;.$$

Thus the summation over all $s$ results in:

$$\sum_{s \in S} Pr(s|\,\text{D'}(s)) Pr(F(x;s) = y|s) = \frac{c(\text{D'} \cup \{(x,y)\})}{c(\text{D'})} \;.$$

As $c(\text{D'})$ is a constant given D' and is invariant under $\operatorname*{argmin}_{(x,y)}$, we have $\operatorname*{argmin}_{(x,y)} c(\text{D'} \cup \{(x,y)\}) = \operatorname*{argmin}_{(x,y)} Pr((x,y)|\,\text{D'})$ as claimed.

DRAWING PROGRAM EXAMPLES

The following images are some examples of what can be synthesized with the drawing program. Each row has the target image on the left. Next to the target image is the observations chosen by the

neural network followed by the neural network's estimation of the image. The recovered parameters from the image are on the right.

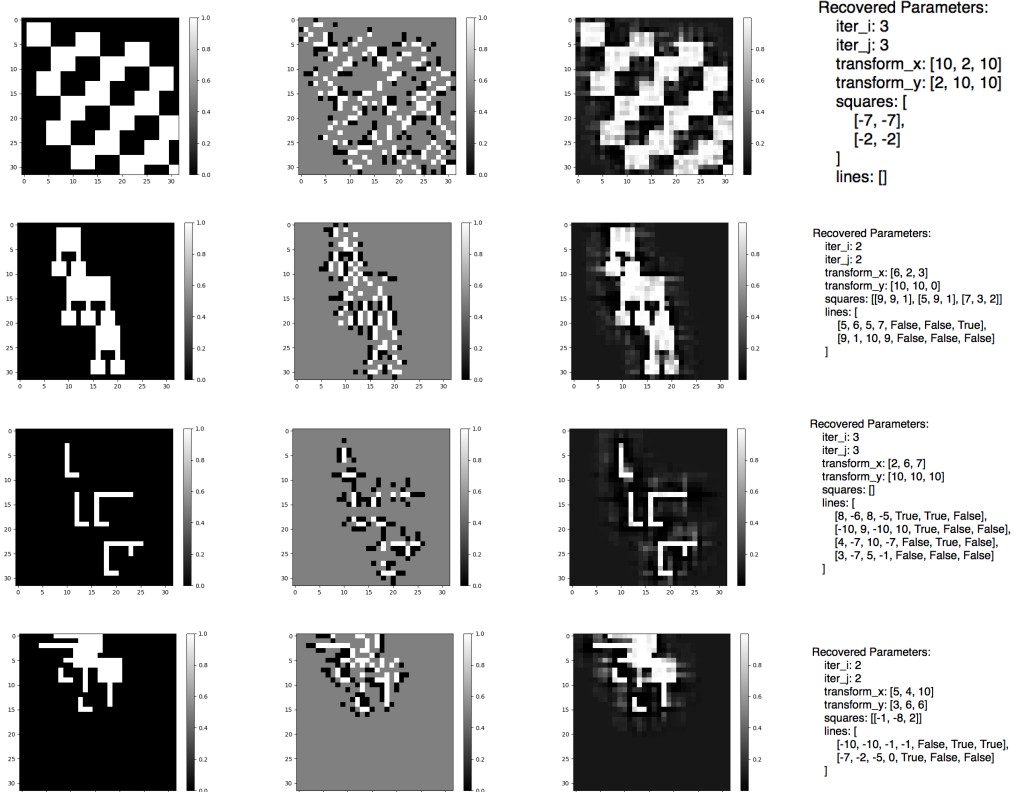

SEQUENCE OF PREDICTION ESTIMATES

The following are a sequence of the neural network's approximation of the render given its current observations. The sampling of observations is shown on the top and the corresponding neural network approximation is shown underneath it.

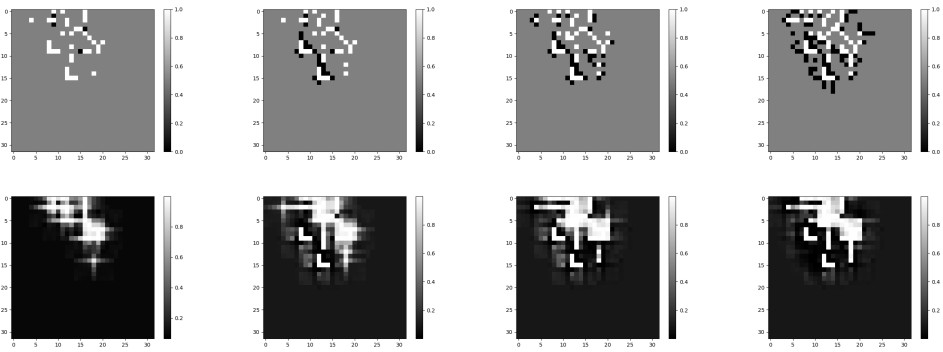

