# OpenReview forum: "Learning to select examples for program synthesis"
_ICLR.cc/2018/Conference — Reject_

### Official Review · AnonReviewer3 · 2017-11-20
**Interesting work, but underwhelming empirical evaluation.**

**Rating:** 4
**Confidence:** 4

**Review:**

The paper proposes a method for identifying representative examples for program
synthesis to increase the scalability of existing constraint programming
solutions. The authors present their approach and evaluate it empirically.

The proposed approach is interesting, but I feel that the experimental section
does not serve to show its merits for several reasons. First, it does not
demonstrate increased scalability. Only 1024 examples are considered, which is
by no means large. Even then, the authors approach selects the highest number of
examples (figure 4). CEGIS both selects fewer examples and has a shorter median
time for complete synthesis. Intuitively, the authors' method should scale
better, but they fail to show this -- a missed opportunity to make the paper
much more compelling. This is especially true as a more challenging benchmark
could be created very easily by simply scaling up the image.

Second, there is no analysis of the representativeness of the found sets of
constraints. Given that the results are very close to other approaches, it
remains unclear whether they are simply due to random variations, or whether the
proposed approach actually achieves a non-random improvement.

In addition to my concerns about the experimental evaluation, I have concerns
about the general approach. It is unclear to me that machine learning is the
best approach for modeling and solving this problem. In particular, the
selection probability of any particular example could be estimated through a
heuristic, for example by simply counting the number of neighbouring examples
that have a different color, weighted by whether they are in the set of examples
already, to assess its "borderness", with high values being more important to
achieve a good program. The border pixels are probably sufficient to learn the
program perfectly, and in fact this may be exactly what the neural net is
learning. The above heuristic is obviously specific to the domain, but similar
heuristics could be easily constructed for other domains. I feel that this is
something the authors should at least compare to in the empirical evaluation.

Another concern is that the authors' approach assumes that all parameters have
the same effect. Even for the example the authors give in section 2, it is
unclear that this would be true.

The text says that rand+cegis selects 70% of examples of the proposed approach,
but figure 4 seems to suggest that the numbers are very close -- is this initial
examples only?

Overall the paper appears rushed -- the acknowledgements section is left over
from the template and there is a reference to figure "blah". There are typos and
grammatical mistakes throughout the paper. The reference to "Model counting" is
incomplete.

In summary, I feel that the paper cannot be accepted in its current form.

---

> ### Author Response · Authors · 2018-01-05
> **response 3**
>
>
> Only 1024 examples are considered, which is by no means large.
>
> => Indeed this is not large compared to a standard vision task, but if all are taken together, can be quite significant for the constraint solver to reason with. We believe what you meant by “not large” is in a sense that the entire _input space_ is quite small, and we do intend to address this problem so that the input-outputs are not total in the dataset, but rather a sample of input-output that lives in a much bigger space.
>
>
> Even then, the authors approach selects the highest number of
> examples (figure 4). CEGIS both selects fewer examples and has a shorter median
> time for complete synthesis. Intuitively, the authors' method should scale
> better, but they fail to show this -- a missed opportunity to make the paper
> much more compelling. This is especially true as a more challenging benchmark
> could be created very easily by simply scaling up the image.
>
> => We tuned some weights and have better results (better median time).
>
> https://imgur.com/a/JyZor
>
> It is true that CEGIS selects the fewest number of examples, but it does so at a cost of calling a constraint solver each time. So that whenever a constraint solver is “stuck” the example would not be produced for a long time. Our approach selects a whole bunch of examples in batch at a low cost (using neural network) and often no additional example is required (i.e. no additional solver time is needed to pick more examples). We should scale up the images and see how they compare (originally we had 64x64 images with very good results but it was taking forever to run even 1 instance).
>
>
>
>
> Second, there is no analysis of the representativeness of the found sets of
> constraints. Given that the results are very close to other approaches, it
> remains unclear whether they are simply due to random variations, or whether the
> proposed approach actually achieves a non-random improvement.
>
> => The additional cegis example needed is a quantification of this metric.
>
> https://imgur.com/a/JyZor
>
> Our approach selected examples in a way that the synthesizer returned a correct program with 0 or 1 additional cegis examples on top. Meaning the original set of example chosen by the NN is forcing a set of constraints strong enough so the correctly synthesized program cannot be ambiguous. However, a better metric would be to explicitly measure this ambiguity.
>
>
> The above heuristic is obviously specific to the domain, but similar
> heuristics could be easily constructed for other domains. I feel that this is
> something the authors should at least compare to in the empirical evaluation.
>
> => We will incorporate the boarder heuristic as another baseline to compare against (one issue with this heuristic is all boarders is clearly a lot of input-output examples, do you suggest to keep all of them? Or you stop collecting at some point, and if so what is a good stopping criteria if you intend to do so without any learning but rely on a heuristic?) We will include experiments from other domain(s) such that it will convince the reader that there will be cases where heuristics are hard to construct.
>
>
> Overall: Quantify the "representativeness" of the set of examples better, perhaps explicitly. Incorporate new domains and show that learning to select examples is more reasonable than hacking a heuristic for each domain.

---

### Official Review · AnonReviewer1 · 2017-11-27
**Interesting formulation, but execution lets the paper down**

**Rating:** 5
**Confidence:** 4

**Review:**

This paper presents a method for choosing a subset of examples on which to run a constraint solver
in order to solve program synthesis problems. This problem is basically active learning for
programming by example, but the considerations are slightly different than in standard active
learning. The assumption here is that labels (aka outputs) are easily available for all possible
inputs, but we don't want to give a constraint solver all the input-output examples, because it will
slow down the solver's execution.

The main baseline technique CEGIS (counterexample-guided inductive synthesis) addresses this problem
by starting with a small set of examples, solving a constraint problem to get a hypothesis program,
then looking for "counterexamples" where the hypothesis program is incorrect.

This paper instead proposes to learn a surrogate function for choosing which examples to select. The
paper isn't presented in exactly these terms, but the idea is to consider a uniform distribution
over programs and a zero-one likelihood for input-output examples (so observations of I/O examples
just eliminate inconsistent programs). We can then compute a posterior distribution over programs
and form a predictive distribution over the output for all the remaining possible inputs. The paper
suggests always adding the I/O example that is least likely under this predictive distribution
(i.e., the one that is most "surprising").

Forming the predictive distribution explicitly is intractable, so the paper suggests training a
neural net to map from a subset of inputs to the predictive distribution over outputs.  Results show
that the approach is a bit faster than CEGIS in a synthetic drawing domain.

The paper starts off strong. There is a start at an interesting idea here, and I appreciate the
thorough treatment of the background, including CEGIS and submodularity as a motivation for doing
greedy active learning, although I'd also appreciate a discussion of relationships between this approach
and what is done in the active learning literature.Once getting into the details of the proposed approach,
the quality takes a downturn, unfortunately.

Main issues:
- It's not generally scalable to build a neural network whose size scales with the number
of possible inputs. I can't see how this approach would be tractable in more standard program
synthesis domains where inputs might be lists of arrays or strings, for example.  It seems that this
approach only works due to the peculiarities of the formulation of the only task that is considered,
in which the program maps a pixel location in 32x32 images to a binary value.

- It's odd to write "we do not suggest a specific neural network architecture for the
middle layers, one should seelect whichever architecture that is appropriate for the domain at
hand." Not only is it impossible to reproduce a paper without any architectural details, but the
result is then that Fig 3 essentially says inputs -> "magic" -> outputs. Given that I don't even
think the representation of inputs and outputs is practical in general, I don't see what the
contribution is here.

- This paper is poor in the reproducibility category. The architecture is never described,
it is light on details of the training objective, it's not entirely clear what the DSL used in the
experiments is (is Figure 1 the DSL used in experiments), and it's not totally clear how the random
images were generated (I assume values for the holes in Figure 1 were sampled from some
distribution, and then the program was executed to generate the data?).

- Experiments are only presented in one domain, and it has some peculiarities relative to
more standard program synthesis tasks (e.g., it's tractable to enumerate all possible inputs).  It'd
be stronger if the approach could also be demonstrated in another domain.

- Technical point: it's not clear to me that the training procedure as described is consistent
with the desired objective in sec 3.3. Question for the authors: in the limit of infinite training
data and model capacity, will the neural network training lead to a model that will reproduce the
probabilities in 3.3?

Typos:
- The paper needs a cleanup pass for grammar, typos, and remnants like "Figure blah shows our
neural network architecture" on page 5.

Overall: There's the start of an interesting idea here, but I don't think the quality is high enough
to warrant publication at this time.

---

> ### Author Response · Authors · 2018-01-05
> **response 2**
>
> I'd also appreciate a discussion of relationships between this approach and what is done in the active learning literature.
>
> => Which work would you see as most similar to our work? I am seeing CEGIS most closely relates to the line of work that ask for labels for the input that lies most "close to the decision boundary" for learning a SVM. However I am in a setting where all labels are already given but are too many to process. If you can give a few pointers/papers on what would be good related work in this space it would be very well appreciated.
>
>
> It's not generally scalable to build a neural network whose size scales with the number
> of possible inputs. I can't see how this approach would be tractable in more standard program
> synthesis domains where inputs might be lists of arrays or strings, for example.  It seems that this
> approach only works due to the peculiarities of the formulation of the only task that is considered,
> in which the program maps a pixel location in 32x32 images to a binary value.
>
> => You are right. In the particular experiments we use a conv-net of a 7x7 window size so it would scale to arbitrary large images (to the point that the constraint synthesizer is the bottleneck). However in general it is definitely true such encoding will not scale. We are working on a rnn architecture that do not take in the entire input space at once.
>
>
> - This paper is poor in the reproducibility category. The architecture is never described,
> it is light on details of the training objective, it's not entirely clear what the DSL used in the
> experiments is (is Figure 1 the DSL used in experiments), and it's not totally clear how the random
> images were generated (I assume values for the holes in Figure 1 were sampled from some
> distribution, and then the program was executed to generate the data?).
>
> => We'll do a better job next time explaining the architecture and the DSL. The random images are generated by uniformly sampling inter values (between some range bounds) on the wholes in Figure 1, and the draw program is executed to generate a 32x32 image.
>
>
> - Experiments are only presented in one domain, and it has some peculiarities relative to
> more standard program synthesis tasks (e.g., it's tractable to enumerate all possible inputs).  It'd
> be stronger if the approach could also be demonstrated in another domain.
>
> => We do intend to take our work to a different domain and have some in mind. However, if you have any domain where you would like to see us try this approach please let us know, it would be very instructive.
>
>
> - Technical point: it's not clear to me that the training procedure as described is consistent
> with the desired objective in sec 3.3. Question for the authors: in the limit of infinite training
> data and model capacity, will the neural network training lead to a model that will reproduce the
> probabilities in 3.3?
>
> => Yes it will. The neural network in that case would act like a "soft" dictionary of counts keeping track of all the instances a new input x is mapped to y conditioned on all the past observed input/outputs. Thus, for the same reason the explicit count formulation approaches the desired probability, the neural network would as well.
>
>
> Overall: Need better explaination on the neural network architecture, a new domain is needed (with a better architecture that can scale)

---

### Official Review · AnonReviewer2 · 2017-11-30
**Good idea, but some misgivings about accepting in current state.**

**Rating:** 5
**Confidence:** 3

**Review:**

General-purpose program synthesizers are powerful but often slow, so work that investigates means to speed them up is very much welcome—this paper included. The idea proposed (learning a selection strategy for choosing a subset of synthesis examples) is good. For the most paper, the paper is clearly-written, with each design decision justified and rigorously specified. The experiments show that the proposed algorithm allows a synthesizer to do a better job of reliably finding a solution in a short amount of time (though the effect is somewhat small).

I do have some serious questions/concerns about this method:

Part of the motivation for this paper is the goal of scaling to very large sets of examples. The proposed neural net setup is an autoencoder whose input/output size is proportional to the size of the program input domain. How large can this be expected to scale (a few thousand)?

The paper did not specify how often the neural net must be trained. Must it be trained for each new synthesis problem? If so, the training time becomes extremely important (and should be included in the “NN Phase” time measurements in Figure 4). If this takes longer than synthesis, it defeats the purpose of using this method in the first place.
Alternatively, can the network be trained once for a domain, and then used for every synthesis problem in that domain (i.e. in your experiments, training one net for all possible binary-image-drawing problems)? If so, the training time amortizes to some extent—can you quantify this?
These are all points that require discussion which is currently missing from the paper.

I also think that this method really ought to be evaluated on some other domain(s) in addition to binary image drawing. The paper is not an application paper about inferring drawing programs from images; rather, it proposes a general-purpose method for program synthesis example selection. As such, it ought to be evaluated on other types of problems to demonstrate this generality. Nothing about the proposed method (e.g. the neural net setup) is specific to images, so this seems quite readily doable.

Overall: I like the idea this paper proposes, but I have some misgivings about accepting it in its current state.




What follows are comments on specific parts of the paper:


In a couple of places early in the paper, you mention that the neural net computes “the probability” of examples. The probability of what? This was totally unclear until fairly deep into Section 3.
 - Page 2: “the neural network computes the probability for other examples not in the subset”
 - Page 3: “the probability of all the examples conditioned on…”

On a related note, I don’t like the term “Selection Probability” for the quantity it describes. This quantity is ‘the probability of an input being assigned the correct output.’ That happens to be (as you’ve proven) a good measure by which to select examples for the synthesizer. The first property (correctness) is a more essential property of this quantity, rather than the second (appropriateness as an example selection measure).

Page 5: “Figure blah shows our neural network architecture” - missing reference to Figure 3.

Page 5: “note that we do not suggest a specific neural network architecture for the middle layers, one should select whichever architecture that is appropriate for the domain at hand” - such as? What are some architectures that might be appropriate for different domains? What architecture did you use in your experiments?

The description of the neural net in Section 3.3 (bottom of page 5) is hard to follow on first read-through. It would be better to lead with some high-level intuition about what the network is supposed to do before diving into the details of how it’s set up. The first sentence on page 6 gives this intuition; this should come much earlier.

Page 5: “a feed-forward auto-encoder with N input neurons…” Previously, N was defined as the size of the input domain. Does this mean that the network can only be trained when a complete set of input-output examples is available (i.e. outputs for all possible inputs in the domain)? Or is it fine to have an incomplete example set?

---

> ### Author Response · Authors · 2018-01-05
> **response 1**
>
> Part of the motivation for this paper is the goal of scaling to very large sets of examples. The proposed neural net setup is an autoencoder whose input/output size is proportional to the size of the program input domain. How large can this be expected to scale (a few thousand)?
>
> => This is a fair point. We also believe the current architecture is both badly explained and badly constructed for other kind of tasks. We failed to mention that the particular architecture for the drawing example is a conv-net with a 7x7 window size, so there is an additional independence assumption based on location: pixel values far away from each other are uncorrelated. For that particular task local informations such as the shape of a line / square is already sufficient for picking good examples for synthesis, and scales well potentially to very large images. We also hope to include experiments on a textual domain in the future, which will use a recurrent neural network architecture that sequentially process the input-output rather than all at once.
>
>
>
> The paper did not specify how often the neural net must be trained. Must it be trained for each new synthesis problem?
>
> => It is trained once as a kind of “compilation” if you will for a domain. Once trained it can be used repeatedly without additional training.
>
>
> I also think that this method really ought to be evaluated on some other domain(s) in addition to binary image drawing.
>
> => Indeed! We really hoped for it too but could not quite get it working in the time of the deadline. We agree that a general-purpose paper would benefit from a additional domains.
>
>
> Overall the specific neural network architectures need to be better explained, with potentially a different architecture for a different domain to show that it can scale to potentially large input-spaces. We will take these suggestions to make the work more solid. Thanks!

---

### Decision · Program_Chairs · 2018-01-29
**ICLR 2018 Conference Acceptance Decision**

**Decision:**

Reject

**Comment:**

The reviewers were largely agreed that the paper presented an interesting idea and has potential but needs a better empirical evaluation.  It seems that the authors largely agree and are working to improve it.

PROS:
1. Improving the speed of program synthesis is a useful problem
2. Good treatment of related work, e.g. CEGIS

CONS:
1. The approach likely does not scale
2. The architecture is underspecified making it hard to reproduce
3. Only 1 domain for evaluation